# Implementation and Characterization of a Laminate Hybrid Composite Based on Palm Tree and Glass Fibers

**DOI:** 10.3390/polym13193444

**Published:** 2021-10-08

**Authors:** Hamid Kaddami, Oumaima Hafs, Taha EL Assimi, Lamia Boulafrouh, El-Houssaine Ablouh, Mohamed Mansori, Hicham Banouni, Said Bouzit, Fouad Erchiqui, Khalid Benmoussa, Fatima-ezzahra Arrakhiz

**Affiliations:** 1Laboratory of Innovative Materials for Energy and Sustanable Development (IMAD-Lab), Faculty of Sciences and Technologies, Cadi Ayyad University, Marrakech 40000, Morocco; oumaimahafs@gmail.com (O.H.); tahaensmarrakech@gmail.com (T.E.A.); lamia.boulafrouh@gmail.com (L.B.); m.mansori@uca.ac.ma (M.M.); 2Institute of Science, Technology & Innovation, Mohammed VI Polytechnic University, Lot 660, Hay Moulay Rachid, Ben Guerir 43150, Morocco; lhoussainiblah@gmail.com; 3Laboratory of Metrology and Information Processing, Ibn Zohr University, Agadir 80000, Morocco; h.banouni@uiz.ac.ma; 4Laboratoire de Thermodynamique et Énergétique, Faculty of Science, Ibn Zohr University, Cité Dakhla, B.P. 8106, Agadir 80000, Morocco; saidbouzit19@gmail.com; 5Laboratory of Biomaterials, Université du Québec en Abitibi-Témiscamingue, 445 Boulevard de l’Université, Rouyn-Noranda, QC J9X 5E4, Canada; 6Laboratoire des Sciences de L’ingénieur Faculty of Science, Ibn Zohr University, Cité Dakhla, B.P. 8106, Agadir 80000, Morocco; k.benmoussa@uiz.ac.ma; 7Laboratory of Electronics Signal Processing and Physical Modeling, Faculty of Science, Ibn Zohr University, Cité Dakhla, B.P. 8106, Agadir 80000, Morocco

**Keywords:** RTM, hybrid composite, natural fibers, glass fibers, mechanical properties, ultrasonic waves, aging

## Abstract

In this work, laminated polyester thermoset composites based on palm tree fibers extracted from palms leaflets and glass mats fibers were manufactured to develop hybrid compositions with good mechanical properties; the mixture of fibers was elaborated to not exceed 25 vol.%. Samples were prepared with a resin transfer molding (RTM) method and mechanically characterized using tensile and flexural, hardness, and impact tests, and ultrasonic waves as a non-destructive technique. The water sorption of these composite materials was carried out in addition to solar irradiation aging for approximately 300 days to predict the applicability and the long-term performance of the manufactured composites. Results have shown that the use of glass fibers significantly increased all properties; however, an optimum combination of the mixture could be interesting and could be developed with less glass sheet and more natural fibers, which is the goal of this study. On the other hand, exposure to natural sunlight deteriorated the mechanical resistance of the neat resin after only 60 days, while the composites kept high mechanical resistance for 365 days of exposure.

## 1. Introduction

Nowadays, natural-fibers-reinforced polymer composites have greatly increased in exterior applications. The two-dimensional structures have been used in maritime craft, aircraft, high-performance automobiles, and civil infrastructure [1,2,3,4,5,6]. Some of the advantages of natural fibers are their exceptional properties compared to synthetic ones, such as low cost, recyclability, renewability, and good mechanical properties [7]. Despite their use as short reinforcement in the polymer matrix, long fiber sheets in composites present several advantages when they provide inherent reinforcement in multiple directions [8,9,10], especially when they are used with organic polymer matrices. Such studies encourage the use of natural fibers in hybrid composites with synthetic fibers, and this combination has a good effect on the development of mechanical proprieties [11,12].

However, to achieve such required objectives, a good knowledge of composite materials is needed. Additionally, the choice of reinforcements and the matrix binder, and the process of composites manufacturing depends on the known fibers’ morphology, comportment under stress, and environment [13].

In this work, a hybrid polymeric composite was made with date palm fibers and glass fibers sheets. Composites were manufactured using the resin transfer molding (RTM) method. Natural fibers (NF) layers were alternated with glass fiber (GF) layers up to 25 vol.% fractions. Mechanical characterization tests such as tensile, three-points bending, impact, and hardness tests were carried out to characterize hybrid composites. Additionally, in this study, the ultrasonic waves technique was carried out to calculate the Young’s modulus of the different samples and their Poisson’s ratio. This technique is a non-destructive method to mechanically characterize our hybrid composites [14,15].

The multilayer structure and the natural fiber percentage affect the studied properties. Owing to the differences in the mechanical and viscoelastic properties of the glass and plant fibers, it is predicted that the prepared hybrid composites with different fractions of these two fibers and the same total volume fraction (25 vol.%) will have various mechanical behaviors. The viscoelastic behavior of plant fibers is due to the combination and presence of amorphous polymer (lignin and hemicellulose) and highly crystalline polymer (cellulose) [16]. Composites were also evaluated by water absorption technique to see their resistance to humidity and moisture absorption. Finally, the mechanical changes, for instance, the evolution of flexural modulus, strength, Young’s moduli, and microhardness, are widely viewed as indicators of aging under solar irradiation.

To the best of the authors’ knowledge, no hybrid composites based on date palm tree/glass fibers have been studied, especially with a polyester thermoset resin. On the other hand, in the present paper, the originality lies in the combination, at the same time, of: (i) a multitude of characterizations of the structure and the properties of these palm-tree-based composites. A comparative study between non-destructive (ultrasonic method) and a destructive method (tensile tests) to mechanically characterize the composites will be made. (ii) The presentation of a solar aging study to show how the fibers and their nature could improve the solar aging resistance of the polyester thermoset matrix. The stabilizing effect of the fibers will be explored.

## 2. Materials and Methods

### 2.1. Materials

Polyester resin (PEs) was used to prepare reinforced composites, and in order to take advantage of local crops, raw date palm fibers (Chamaeropshumilis) provided from Marrakech region were harvested at the end of maturity to be used as reinforcement with commercial glass fibers mat. Both fibers were arranged sheet by sheet to reach the 25 vol.% and the required thickness of the RTM mold. In the glass and date palm tree fibers, mats the fibers are randomly oriented.

### 2.2. Preparation of Laminated Composites

In the present study, the implementation process used for manufacturing the composites is injection RTM (resin transfer molding) in which a liquid pre-catalyzed resin is injected under pressure (2.5 bars, at room temperature) through a fibrous reinforcement positioned in a closed homemade mold, and Figure 1 shows the four steps of RTM process. The polyester mixtures were cross-linked using 1.5% catalyst-type Methyl Ethyl Ketone Peroxide (MEKP), also commercially named AKPEROX A50. The used reinforcement is the mats of glass fibers and date palm fibers. Prepared plates are dimensioned as: (200 mm × 200 mm × 5 mm). The used natural fibers were cut into fibers of 2–3 cm length and 1–2 mm in diameter. Table 1 shows the different fiber layer compositions in the laminated composites.

## 3. Characterizations Techniques

### 3.1. Scanning Electron Microscopy (SEM)

Microscopic observation of date palm fibers was carried out by (XL30ESEM), and it was used to identify the fibrillose morphology of the fibers and evaluate surface interaction between fibers and the polymer at 25 vol.%. The composites were fractured after having been chilled in nitrogen liquid. Fractured surfaces were sputter-coated with gold prior to SEM observation.

### 3.2. ATR–FTIR Analysis

Fourier Transform-Infrared spectra were recorded on an ABB Bomem FTLA 2000-102 spectrometer (using SPECAC GOLDEN GATE). ATR accessory with a resolution of 4 cm^−1^ was used.

### 3.3. Mechanical Testing

#### 3.3.1. Tensile Test

Tensile tests of three specimens from each type of composites (period 0) were performed as rectangular shape, and the average values were reported in terms of the Young’s modulus and tensile strength. The tensile tests were performed on a universal testing machine INSTRON 8821S (Instron, Norwood, MA, USA) [17] at a crosshead speed of 3 mm/min using a 5 kN load cell.

#### 3.3.2. Flexural Test

The three-point bending test was completed using a universal testing machine, INSTRON 3369 (Instron, USA), equipped with a data acquisition unit. Samples were prepared according to ASTM D790-03 [18] to be sized 200 mm × 150 mm × 5 mm (length × width × thickness); for each composite, an average of four specimens were tested. Flexural modulus and strength were evaluated under room-temperature conditions (25 °C, 40% relative humidity).

#### 3.3.3. Impact Testing

Notched Charpy impact strength was measured on a Tinius Olsen machine model Impact IT 504 using a 15 J hammer. The rectangular specimens were previously U-notched. Three specimens were tested to obtain an average and standard deviation.

#### 3.3.4. Hardness Test

Hardness test was carried out using Ibertest Universal Hardness Tester, manually loaded. A pre-load was normally applied to the surface of the prepared samples (dimension: 30 × 30 × 5 mm^3^) according to ASTM D785 [19]. A total of four specimens were tested for each composite’s combination. The used hardness test was Rockwell B hardness.

#### 3.3.5. Micro-Hardness Test

The micro indentation tests were carried out at ambient temperature using Future-Tech FM-700e, Japan Vickers microhardness tester equipped with a diamond square pyramid, with an included angle of 136° at the tip. A selected load of 5 N was applied for 10 s, and the measurements were performed according to ASTM standard E384 for micro-indentation hardness of materials [19]. For each sample, the microhardness was measured at five different locations in each face to record an average value.

#### 3.3.6. Ultrasonic Measurement Test

In this section, the ultrasound method is used as a non-destructive mechanical characterization to be compared with destructive mechanical tests (tensile test). The founded results are discussed and compared to highlight advantages and drawbacks of each technique. Firstly, for the ultrasonic measurements, the composite samples were immersed in a homemade water tank during the measurement, which did not exceed 5 min to avoid the moisture content in the composites. Secondly, we proceeded to the calculation of the propagation velocity of the wave in a sample by using flight time measurements of the first two echoes reflected by the walls of the sample. Figure 2 illustrates the main used material for this technique.

Using the pulse echo method [20], it is possible to find ultrasonic properties. Moreover, these techniques provide the elastic properties of Young’s modulus and Poisson’s ratio of the material. The Young’s modulus and Poisson’s ratio from ultrasonic measurement function of *ρ*, the density, and the *V_T_*, *V_L_* are given as [20]:(1)$$E_{u}=\rho V_{T}^{2}\frac{V_{L}^{2}-2V_{T}^{2}}{V_{L}^{2}-V_{T}^{2}}$$
(2)$$v_{u}=\frac{1}{2}\frac{V_{L}^{2}-2V_{T}^{2}}{V_{L}^{2}-V_{T}^{2}}$$
where $V_{L}$ and $V_{T}$ are the longitudinal and transversal waves’ velocities, respectively.

The calculation of the propagation velocity of the wave in a material by flight time measurements is made from the first two echoes reflected by the walls of the sample [20].

The measurement of longitudinal wave velocity corresponds to the incident wave that propagates at normal incidence to the surface of the sample when the transversal wave velocity *V_T_* is measured for an incidence angle θ relative to the normal of the surface of the sample.

The densities of the prepared composites are calculated by the use of the mixture rule given above, taking into account the fibers’ content and the densities of the various constituents of the composites:(3)$$\rho_{th}={\left(\sum_{i}^{}\frac{m_{f,i}}{\rho_{i}}\right)}^{-1}$$
where *m_f,i_* is the mass fraction and *ρ_i_* is the density, where *i* is the reinforcement and the matrix.

#### 3.3.7. Water Uptake Tests

Water uptake tests were conducted by submerging the composite specimens in a water bath at room temperature. The specimens were taken out from water, wiped with filter paper to eliminate surface water, weighted in a high precision balance, and then immersed again in water. This process was repeated every two hours until reaching saturation plateau. Measurement of the specimens’ weights was completed as quickly as possible to avoid evaporation of water effects.

#### 3.3.8. Solar Radiation

For test of solar radiation, the specimens were exposed to natural solar radiation for 400 days (presented as 8 periods), with an average temperature of 28.6 °C from March 2018 to April 2019 in our laboratory at Marrakesh, Morocco, where periods are noted P0, P1, P2, P3, P4, P5, P6, P7, P8, referring to 0, 20, 40, 60, 130, 200, 290, 365, and 400 days.

## 4. Results and Discussion

### 4.1. Structural Characteristics of the Date Palm Fibers

Figure 3b,c shows the longitudinal and transverse morphological structure of date palm fiber, respectively. These photos clearly show the main components of the leaflets of the date palm tree, viz. parenchyma cells and fibers (dead cells or mature cells, or tracheids or sclerenchyma fibers). More detail about the structure of date palm tree fibers can be found in [21]. The FTIR spectra of palm date fibers are presented in Figure 3d, and the infrared absorption peaks related to the main functional groups of the date palm fibers are found in Table 2. It is clear from the table that date palm fiber contains the main functional groups of lignocellulosic fibers: lignin, cellulose, and hemicellulose. For example, the peak at 1724 cm^−1^ corresponding to hemicelluloses that were identified in previous work [21]. The lignin and hemicelluloses could be removed by alkali treatment of natural fibers to improve fiber/matrix interfacial adhesion [22].

### 4.2. Mechanical Properties

#### 4.2.1. Stiffness of the Composites Obtained from Ultrasonic and Tensile Measurements

In this section, the ultrasound and mechanical measurements are presented and compared as possible characterization methods. Table 3 shows the evolution of both Young’s modulus and tensile strength values as a function of fibers loading. It is also clear from Table 3 that the addition of both fillers enhances the stiffness of the manufactured material. The noted Young’s modulus of 25 wt.% of natural fibers composites is slightly higher than the neat resin (3 GPa) and increases with the addition of glass fibers’ charge against natural fibers. Thus, the obtained gain was 40% at 25 wt.% natural fibers charge, and 133% gain at 25 wt.% glass fibers charge compared to neat resin when 20% GF/5% NF and 10% GF/15% NF hybrid composites reached a gain of 100% and 78.3%, respectively, from the neat resin. This is normal following the high mechanical properties of glass fibers that are stiff compared to cellulosic ones that are viscoelastic [26]. It is interesting to mention that the viscoelastic character of natural fibers is mainly due to the polymeric nature of its constituents, such as cellulose, lignin, and hemicellulose [26]. Thus, the use of both charges as rigid fillers in polymer composites enhances their rigidity since the rigidity of the inorganic fibers is still higher in general compared to the organic material. Moreover, the tensile strength presents a significant increase with the addition of glass fibers while the use of just natural fibers decreases the tensile strength by 42%, which is probably due to the non-adhesion between the resin and natural fibers; moreover, a visual exam demonstrates that fibers’ wettability with resin is poor. The calculated values for Young’s modulus with the ultrasonic technique are nearly the same as those obtained from the classical Young’s modulus, as seen in Table 3, except for the resin specimen, where results are different, probably due to the presence of air bubbles, considering that studying the propagation of ultrasonic waves into the material also allows relating the damage to the anisotropy of the response of the material [27]. Thus, the ultrasound measurements show that the Young’s moduli increase firstly with the fiber content and with glass fibers’ proportion in the hybrid composites.

The ultrasound measurements (Table 3) also allowed us to calculate Poisson’s coefficient of manufactured composites. As a result, the variations of Poisson’s ratio as a function of fibers content are not really significant, which is supported by other studies [20].

These conclusions allow us to use the ultrasonic technique to evaluate the mechanical properties of the same composites under solar irradiation at the bottom of this study.

#### 4.2.2. Flexural Properties

In general, mechanical properties of composites are governed by parameters such as: the architecture of the layers, yarn size, yarn spacing length, angle of orientation, fiber volume fraction, stacking orientation, etc. The curve in Figure 4 illustrates the three-points bonding flexural test measurement carried out using an Instron testing machine. From the stress–strain curve, flexural modulus and flexural stress were measured following Equations (4) and (5) below [28,29]:(4)$$E=\frac{S^{3}}{4\;bh^{3}}m\;$$
(5)$$\;\sigma_{f}=\frac{3PS}{2bh^{2}}$$
where

*P*: applied load (N); *E*: flexural modulus; *S*: support span; *bh*: Width and thickness of the tested sample; *σ_f_*: Flexural stress and *m*: The gradient of the initial straight-line portion of the load–deflection curve.

**Figure 4 polymers-13-03444-f004:**
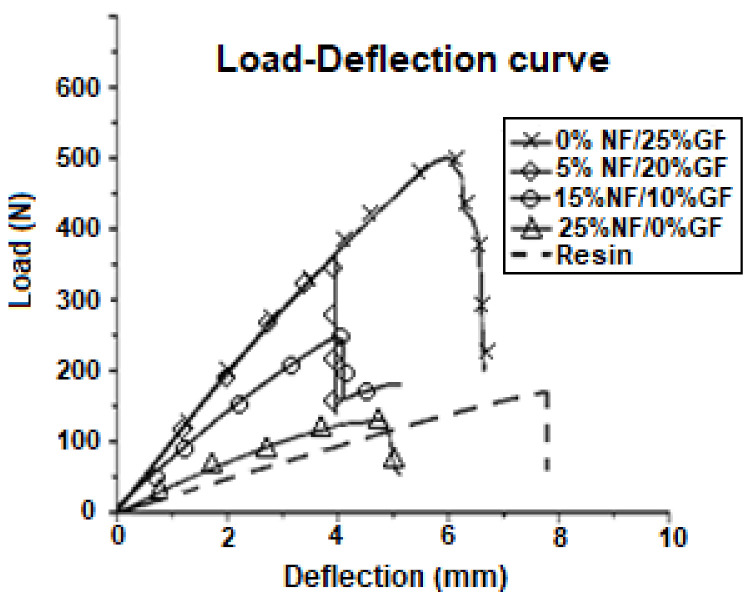
Load–deflection curve of the three-points bonding flexural test.

Table 4 summarizes the evolution of flexural modulus and flexural stress with varying fiber fractions. It is clear from the table and the load–deflection curves that the flexural modulus and load at break decreased with increasing natural fiber content. Additionally, it is clear that bending stress is high for composite reinforced with 25% glass fiber to decrease with increasing the percentage of lignocellulosic fiber in the composite. However, for a fraction of 5 vol.% natural fiber, the composite has good mechanical properties. From these results, it is found that flexural properties are closely dependent on the natural fiber fraction, so the composite became more brittle and less efficient with a high natural fiber percentage. This is probably due to the low wettability of the fibers with the polymer matrix [28].

On the other hand, 5 vol.% of glass fiber can be replaced by natural fiber without penalizing the mechanical performance of the composite with 25% glass fibers.

#### 4.2.3. Impact Properties

The results presented in Table 5 showed that the variation of impact energy depends on the rate of date palm fibers; according to the table data, the impact energy decreases with increasing date palm fiber content up to 25 wt.%. This decrease can be attributed to the insufficient interfacial adhesion between the resin and date palm fibers, as the volume percentage of fibers increases with the number of plies, as well as the percentage of porosity.

The proportion of the resin bonding layer decreases with respect to the entire composite as the number of plies increases. Additionally, the brittle character of natural fiber leads to a weak resistance to the applied load than the composites with glass fibers [30].

#### 4.2.4. Hardness Properties

The variation of the Rockwell hardness value (RH) for laminate composites is shown in Table 6. The hardness of composite materials increases with an increase in glass fiber content up to 25 vol.%. The results showed that GF had a good ability to endure plastic deformation more than natural fibers.

#### 4.2.5. Water Uptake Properties

The obtained results are presented in Figure 5. The presence of natural fibers increases the water absorption, so samples with a high natural fiber percentage have the highest absorption. Water is rapidly absorbed during the first hours, reaching a saturation point, then stabilizes. The diffusion starts to decrease when glass fibers are substituted for date palm layers in the composites. The same curve shows that the diffusion is still invariant at 25 vol.% of glass fiber compared to the resin. These results are explained by the hydrophilicity of cellulosic fibers. Here, date palm fibers interact with water at the surface and in the bulk in contrast to glass fibers that just have surface water absorption [31].

Additionally, the used polyester resin has a different chemical structure compared to the natural fibers owing to poor adhesion, which leads to voids and porosity at the matrix/fiber interface. These problems could be overcome by treating fiber surfaces to be less hydrophilic.

The same results are reached in other studies for different natural fibers composites [32]. Additionally, this study provides an overview of the effect of treatment to reduce the hydrophilic nature of cellulosic fibers, as water absorption study is one of the main factors that affect the mechanical properties of the used composites [33,34,35].

The diffusion coefficient could be calculated using the curves data in Figure 6, following Equation (6):(6)mf+mimi=2LDπ12.t12
where *L* is the thickness of the studied sample and *m_i_* and *m_f_* are, respectively, initial and final weight at time *t*.

Figure 6 summarizes the evolution of the diffusion coefficient (D) vs. date palm fibers percentage. From the curve, the diffusion coefficient is closely related to the nature of the layer. It is increasing with the lignocellulosic fibers percentage, which is normal in view of the hydrophilic nature of this charge.

### 4.3. Composite Behavior during Solar Aging

#### 4.3.1. Mechanical Results

Figure 7 and Figure 8 present, respectively, the evolutions of the flexural modulus and the flexural strength of hybrid composites during one year of solar irradiation, from March 2018 to March 2019.

When comparing the results of this study (Figure 7 and Figure 8), it is clear that the mechanical properties, in terms of flexural and strength moduli of the different hybrid materials, fluctuate around average values during this solar aging period of 290 days, except for the last period, where it is clear that hybrids are strongly affected by the solar irradiation in flexion. Additionally, it is clear from Figure 9, which exposes the Young’s moduli of aged composites using the ultrasonic method, that the rigidity of whole composites is not affected by the solar irradiation but steel affected by the nature of the used fibers.

Indeed, there is not a visible change or aging phenomenon in the mechanical properties. These results indicate a good resistance of the manufactured composites to solar irradiation during this exposure period. Overall, based on these results, the investigated composites material does not seem to be affected by natural solar aging in terms of mechanical properties.

The same figure demonstrates that there are not-marked values, which are explained by the presence of bulks in some samples that affect our tests.

#### 4.3.2. Surface Characterization of Materials

We characterized the mechanical resistance of the surface under an imposed load (5N) to evaluate the solar-aging impact on the resin surface. The Vickers microhardness technique was carried out to composites at four separated aging periods (P0, P3, P5, P7). For these microhardness measurements, at least five different locations for each one of the examined samples were tested, and the average values were recorded in Table 7.

This table shows that the microhardness of the composites increased from 30.8 HV to 34.6 HV after the addition of 25% glass fibers to the matrix and 32.2 HV when the matrix was reinforced with 25% natural fibers. It is clear that the microhardness increased with the addition of fibers, as the polyester resin has low hardness (30.8 HV) at Period 0. Additionally, the reading values for hybrid samples as 20 GF/5% NF and 10% GF/15% NF show that microhardness increased with glass fibers’ charge against natural fibers ones, so 32.7 HV and 31.9 HV were found, respectively, for these hybrids.

However, these microhardness values were noticeably decreased when the samples were exposed to solar irradiation, and it can be seen from sample surfaces (Figure 10) that the surfaces are somehow affected by the solar irradiation. It was found that the 25% NF specimens (P7) were 5% lower in microhardness than 25% NF specimens (P0), while at the same time, 25% GF specimens (P7) were 12% lower than 25% GF specimens (P0), even though hybrids 20% GF/5% NF and 10% GF/15% NF noted, under the same conditions, a decrease of 6.5% and 8.7%, respectively, compared to Period 0 (P0).

Figure 11 shows ATR–FTIR analyses of the exposed surface of the neat matrix and the 10% GF, 15% NF composite. The ATR–FTIR analyses of the other materials are presented in supporting data (S1). One can observe significant changes in the wave number range between 800 and 2000 cm^−1^. These correspond to the broadening of the bands around 1740 cm^−1^ corresponding to the ester groups and their shift toward lower wave numbers. On the other hand, the bands around 1000–1200 cm^−1^, corresponding to the C-C and C-O stretching, show significant changes in their intensities, especially in the composites containing plant fibers. These evolutions might be attributed to the oxidation of the exposed surface, which generates new carbonyl groups (carboxylic acid and aldehyde groups) by alcohol groups’ oxidation and chains scission.

## 5. Conclusions

In this work, hybrid composites were manufactured using RTM processing. The charge fraction in the composites was fixed at 25 vol.%, and the composition was controlled by alternating a total of five layers of natural fibers (NF) and glass fiber (GF). Morphological and structural studies were carried out to describe date palm’s fiber character, and two types of mechanical tests have been carried out on the composites: a non-destructive test using ultrasound and a conventional destructive mechanical test. Flexural, impact, and hardness tests have also been completed and have shown that the presence of natural fibers in the composites (GF/NF/resin) reduces its mechanical performance compared to the binary system (GF/resin). However, the substitution of 5 vol.% of GF by NF could be a mechanically interesting material with somewhat good properties. This represents 20% of the amount of fibers used. On the other hand, natural solar aging tests were carried out to evaluate the resistance of the composites to aging. It was shown that the neat matrix has less resistance to natural aging in comparison to the composites. Its mechanical properties changed significantly after 130 days of sun exposure (significant increase of the modulus and significant reduction of the stress at break). However, the composite has not shown clear variation in the mechanical parameters under solar exposure up to 365 days of exposure. This important result shows that the fibers can play a stabilizing effect again UV radiation. However, when studying the exposed surfaces by the Vickers microhardness and the morphological analysis of the samples surfaces, it was clear that the sample surfaces are significantly affected by solar irradiation after only 90 days of exposure. The presence of the fibers might bring two advantages: (i) they might stop degradation inside the material by stopping UV radiation, and (ii) their reinforcing effect can stop the propagation of cracks induced at the surface of the composition by solar irradiation.

## Figures and Tables

**Figure 1 polymers-13-03444-f001:**
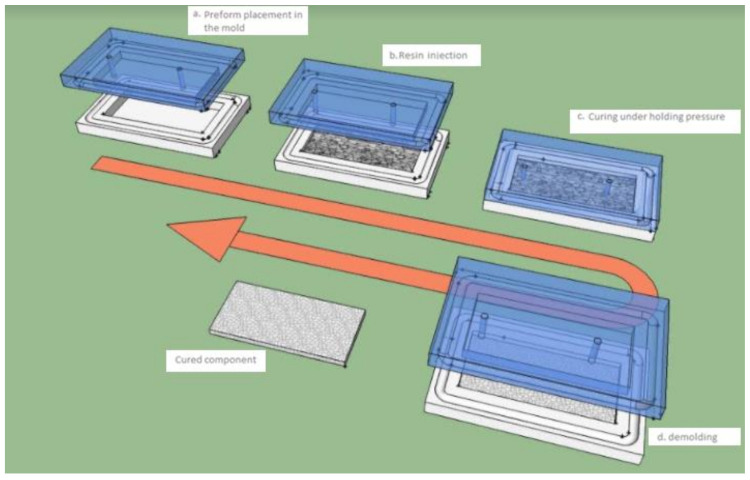
RTM process.

**Figure 2 polymers-13-03444-f002:**
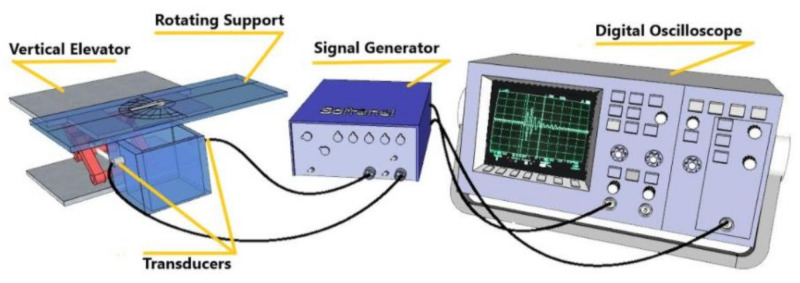
Experimental setup.

**Figure 3 polymers-13-03444-f003:**
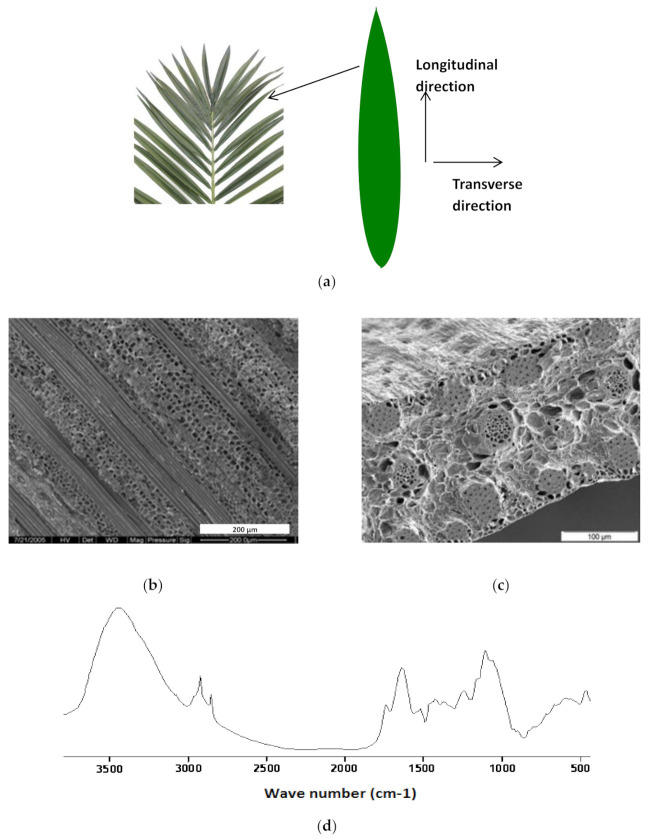
(**a**) Schematic presentation of the date palm tree fibers. (**b**,**c**) Longitudinal and transverse morphological structure of date palm fiber. (**d**) ATR-FTIR spectra of date palm fibers.

**Figure 5 polymers-13-03444-f005:**
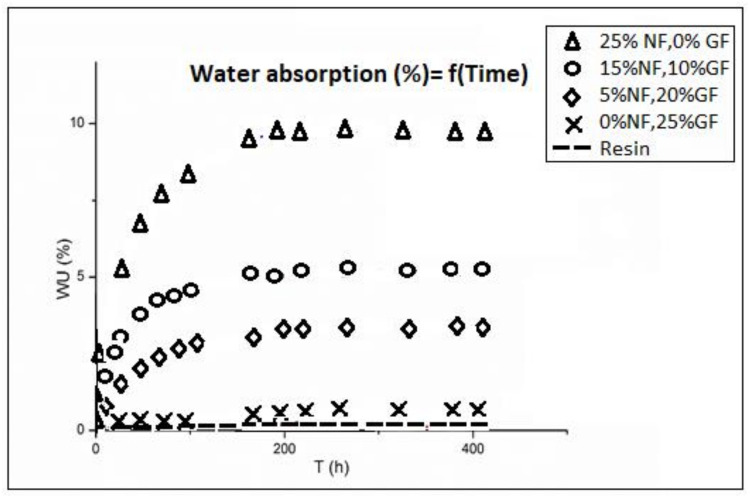
Water absorption vs. time.

**Figure 6 polymers-13-03444-f006:**
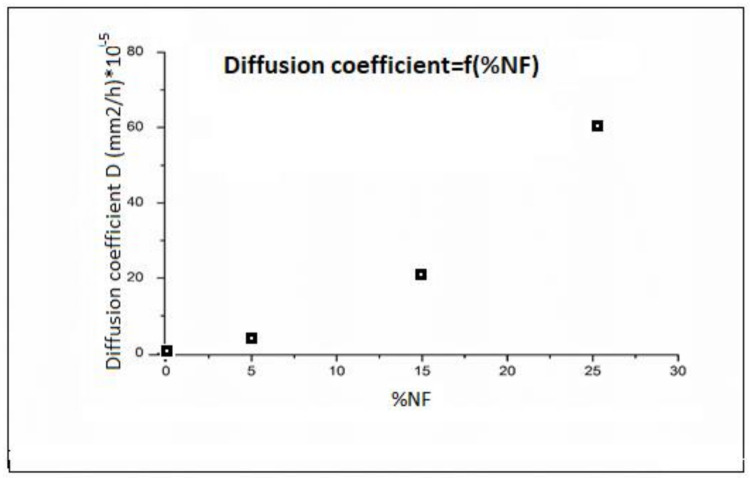
Diffusion coefficient vs. date palm tree fibers volume fraction (%).

**Figure 7 polymers-13-03444-f007:**
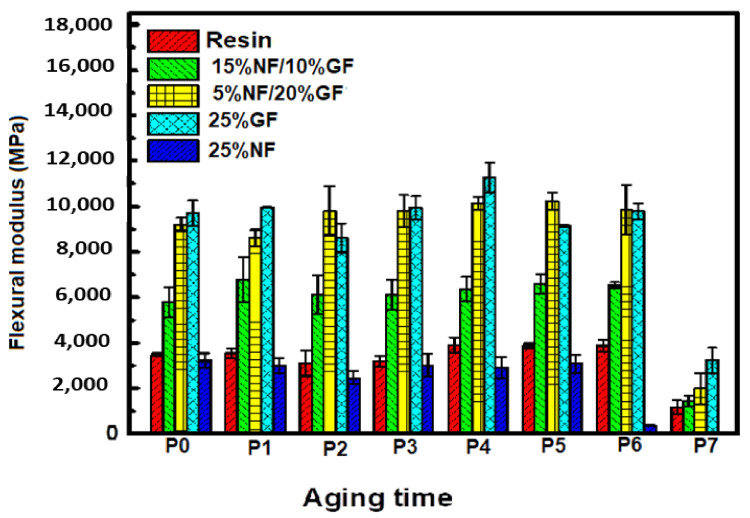
Flexural modulus evolution of hybrid composites during the aging period.

**Figure 8 polymers-13-03444-f008:**
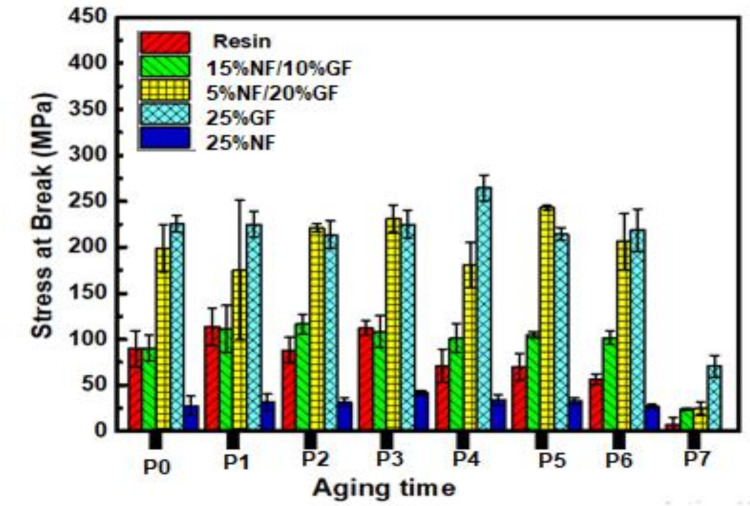
The evolution of flexural stress at break of hybrid composites during the aging period.

**Figure 9 polymers-13-03444-f009:**
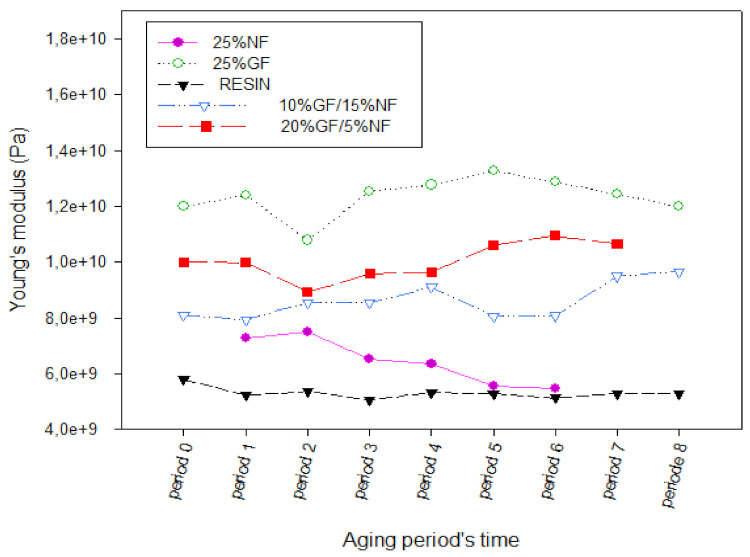
Young’s modulus vs. aging period’s time.

**Figure 10 polymers-13-03444-f010:**
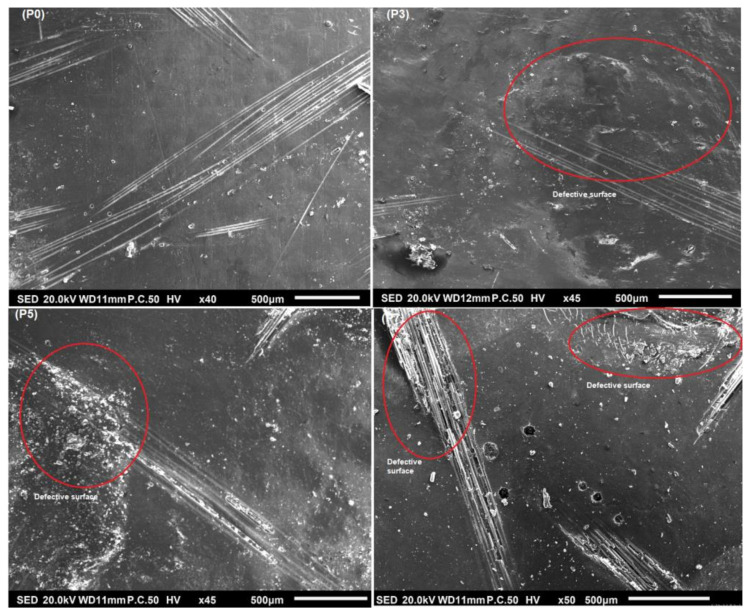
Morphological image of 25% GF composite at different solar irradiation periods (P0, P3, P5, P7) to show the effect of solar irradiation on the sample surfaces.

**Figure 11 polymers-13-03444-f011:**
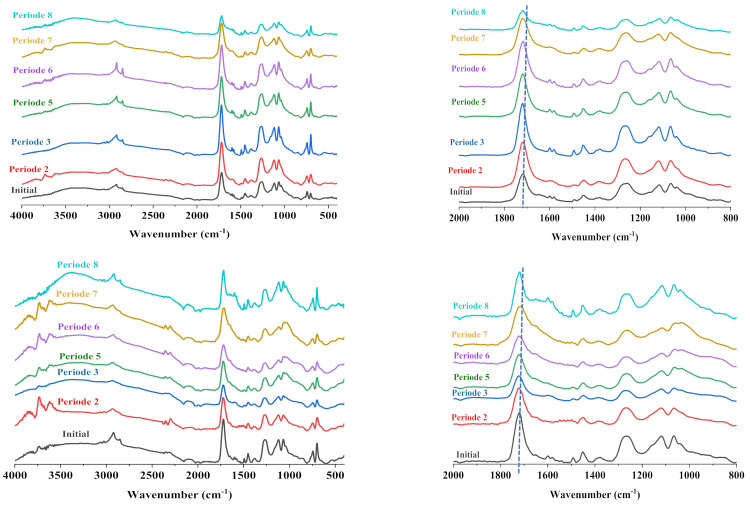
ATR-FTIR spectra of the morphological image of neat resin and 10% GF/15% NF composites at different solar irradiation periods (from P0 to P8).

**Table 1 polymers-13-03444-t001:** The different layer compositions in the laminated hybrid composite.

0% NF/ 25% GF	25% NF/ 0% GF	15% NF/ 10% GF	5% NF/ 20% GF
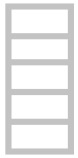		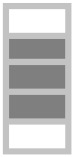	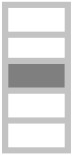
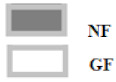			

**Table 2 polymers-13-03444-t002:** Date palm tree fiber FTIR spectrum bands’ attribution.

Wave Number (cm^−1^)	Characteristic Group	Component
3327	-O-H [23]	Alcohol and water
2911	-C-H [24]	Polysaccharide
1724	C=O [23]	Xylan (hemicellulose)
1623	-O-H [23]	Alcohol and water
1506	C=C [23]	Lignin
1424	CH_2_ [23,25]	Polysaccharides
1245	C-C plus O-C plus C=O [23]	Cellulose
1046	C-C, C-OH, C-H [23]	Cellulose, hemicellulose
895	COC, CCO, and CCH [23]	Polysaccharide

**Table 3 polymers-13-03444-t003:** Young’s modulus, tensile strength, and Poisson ratio measurements using a destructive (tensile test) and non-destructive technique (ultrasonic waves).

	Tensile Test Results		Ultrasonic Test Measurements	
	Young’s Modulus (GPa)	Tensile Strength (MPa)	Young’s Modulus (GPa)	Poisson’s Ratio
**Resin matrix**	3	47.4	4.96	0.338804046
**0% NF/25% GF**	4.22	20.1	4.68	0.289017097
**25% GF/0% NF**	7.00	184	7.10	0.354185644
**20% GF/5% NF**	6.02	142	6.73	0.309382566
**10% GF/15% NF**	5.35	82.5	6.55	0.303627805

**Table 4 polymers-13-03444-t004:** Variation of flexural modulus and flexural stress vs. fibers fraction.

Volume Fraction of Fibers Layers	Flexural Stress (MPa)	Flexural Modulus (MPa)
0% GF/25% NF	36 ± 2	2156 ± 148
10% GF/15% NF	75 ± 1	3823 ± 179
20% GF/5% NF	170 ± 17	6583 ± 207
25% GF/0% NF	232 ± 14	8472 ± 302
Resin	75 ± 15	2000 ± 58

**Table 5 polymers-13-03444-t005:** Measured energies at impact testing vs. date palm fibers percentage.

Fiber Volume Fraction	Loss Energy, Eloss (J)	Impact Energy (J) (EAb + Eloss)	Absorption Energy, EAb (J)
**Resin**	4.36	0.14	−4.22
**25% GF**	4.36	0.73	−3.63
**25%NF**	4.36	0.28	−4.08
**10% GF/15% NF**	4.36	0.38	−3.98
**20% GF/5% NF**	4.36	0.5	−3.86

**Table 6 polymers-13-03444-t006:** Variation of hardness by date palm fiber percentage.

Fiber Volume Fraction	Resin	25% GF/0% NF	0% GF/25% NF	10% GF/15% NF	20% GF/5% NF
**RH**	83	96.91	89.16	92.75	92

**Table 7 polymers-13-03444-t007:** Evolution of the Vickers microhardness as function of the solar irradiation period and the fibers loading.

	Aging			
	P0	P3	P5	P7
**Resin**	30.8 HV	30.5 HV	30.6 HV	30.5 HV
**0% GF/25% NF**	32.2 HV	31.2 HV	30.6 HV	30.7 HV
**25% GF/0% NF**	34.6 HV	30.7 HV	30.5 HV	30.5 HV
**20% GF/5% NF**	32.7 HV	30.9 HV	30.6 HV	30.6 HV
**10% GF/15% NF**	31.9 HV	31 HV	30.7 HV	29.1 HV

## Data Availability

The data presented in this study are available on request from the corresponding author.

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
