# Peer review of "Implementation and Characterization of a Laminate Hybrid Composite Based on Palm Tree and Glass Fibers"

_polymers, 2021, doi:10.3390/polym13193444_

Round 1

Reviewer 1 Report

This manuscript discusses the concept of combining glass with wood fibers to make composites and their performance.  The manuscript lacks a discussion regarding the orthotropic nature of the wood fibers and its effect on the composite.  Wood properties within the fiber will depend on whether it was loaded in the longitudinal, tangential or radial direction as per the Forest Products Journal (2021)71 (1): 77–83.  This reference continues to point out that mechanical properties of the wood fiber will depend on the natural polymers such as crystalline cellulose microfibrils and their quantity and orientation, particularly in tensile tests.  As such, wood is fairly heterogenous in direction and underlying makeup while glass fibers is much more homogenous.

Is Fig. 3 a and b the transverse section of wood? 

Crop Fig 3a to look like Fig. 3b and make the white scale error bar so that both images are consistent.  This is very sloppy.

What direction was the ultrasonic test?  Longitudinal to the fiber orientation?  This is why the discussion on the orthotropic behavior of natural fibers is needed as discussed earlier.

Glass fibers are stiff while wood fibers are viscoelastic.  So discussing these differences is important.  Add a reference discussing how lignin, cellulose, and hemicellulose contribute to the viscoelastic response in wood.

Tables have comma’s when they should have decimals.

Author Response

Comments and Suggestions for Authors

Comment 1.1:

This manuscript discusses the concept of combining glass with wood fibers to make composites and their performance.  The manuscript lacks a discussion regarding the orthotropic nature of the wood fibers and its effect on the composite.  Wood properties within the fiber will depend on whether it was loaded in the longitudinal, tangential or radial direction as per the Forest Products Journal (2021)71 (1): 77–83.  This reference continues to point out that mechanical properties of the wood fiber will depend on the natural polymers such as crystalline cellulose microfibrils and their quantity and orientation, particularly in tensile tests.  As such, wood is fairly heterogenous in direction and underlying makeup while glass fibers is much more homogenous.

Response to comment 1.1:

Many thanks to the reviewer to point out this comment.

In this paper the fibers used are date palm tree fibers and glass fibers. Date palm tree fibers are structurally very heterogeneous as this has been clearly discussed and experimentaly proved in our paper (Bendahou, A., Habibi, Y., Kaddami, H., Dufresne, A. Physico-chemical characterization of palm from Phoenix Dactylifera-L, preparation of cellulose whiskers and natural rubber-based nanocomposites. Journal of Biobased Materials and Bioenergy, 2009, 3(1), pp. 81–90) and as this pointed out in the SEM images presented in Fig 3. In fact the characterisations showed that the fibers are composed of long rigid fiber bundle (like wood fibers) embedded in soft cells of sclerite. This makes the question of heterogeneity more complicated than what the respectable reviewer have indicated.

To overcome this problem of high heterogeneity of these composites the authors have prepared composites with many plies of randomly oriented fibers (glass and heterogeneous plant fibers). Sure, as this pointed by the reviewer this is opening a new challenge for the modeling of the properties these composite. At the present time the authors are focusing on the characterisation of the properties as functions of the loads of each fibers, the position of plies in the composites and the effect of solar aging.

To overcome this ambiguity the following sentence has been added in the section 2.1.

‘’ In the glass and date palm tree fibres mats the fibers are randomly oriented’’

Comment 1.2:

Is Fig. 3 a and b the transverse section of wood? 

Response to comment 1.2:

Thank you for the comment. The title of Fig 3 has been changed for more clarification. Schematisation of the longitudinal and transverse directions was added.

Comment 1.3:

Crop Fig 3a to look like Fig. 3b and make the white scale error bar so that both images are consistent.  This is very sloppy.

Response to comment 1.3:

Change wer performed. Clear scale bar have been adde to overcome this ambiguity.  

Comment 1.4:

What direction was the ultrasonic test ? Longitudinal to the fiber orientation?  This is why the discussion on the orthotropic behavior of natural fibers is needed as discussed earlier.

Response to comment 1.4:

Thank you again for this comment. The fibers are randomly oriented. There is no orientation of the fibers. As discussed above and overcome this ambiguity additional sentence has been added in session 2.1.

Comment 1.5:

Glass fibers are stiff while wood fibers are viscoelastic.  So discussing these differences is important.  Add a reference discussing how lignin, cellulose, and hemicellulose contribute to the viscoelastic response in wood.

Response to comment 1.5:

Thank you for this interesting comment. Sur the glass fibers and date palm tree fibers have not the same viscoelastic properties and this will have an effect on the whole physical and mechanical properties of the studied composites. We have added this discussion on the paper and cited the paper suggested by the reviewer. This has been added in two different places of the paper, in the introduction and the discussion of tensile test results. Two references have been added

In the introduction:

Owing to the differences of mechanical and viscoelsatic properties of the glass and plant fibers, the prepared hybrid composites with different fractions of these two fibers and the same total volume fraction (25vol. %) will certainly have various mechanical behavior. The viscoelastic behavior of plant fibers is due to the combination and presence of amorphous polymer (lignin and hemicellulose) and highly crystalline polymer (cellulose) [16].

  1. Zhu Li; Jiali Jiang; Jianxiong Lyu; Jinzhen Cao. Orthotropic Viscoelastic Properties of Chinese Fir Wood Saturated with Water in Frozen and Non-frozen States. Forest Products Journal (2021) 71 (1): 77–83.

In the discussion of tensil test results:

This is normal following the high mechanical properties of Glass fibers that are stiff compared to cellulosic ones that are viscoelastic [26]. It is interesting to mention that the viscoelastic character of natural fibers is mainly due to the polymeric nature of its constituents, as cellulose, lignin, and hemicellulose [26].

  1. cedric montero, these: Caractérisation du comportement viscoélastique asymptotique du bois. Montpellier 2 University (2011).

Comment 1.6:

Tables have comma’s when they should have decimals.

Response to comment 1.6:

Thank you for this comment and sorry for this mistake. Corrections have been made.

Reviewer 2 Report

In this manuscript, the laminated polyester thermoset composites based on palm tree fibers extracted from the palms leaflets and glass mats fibers were manufactured to develop hybrid compositions. However, many researches regarding palm fiber/glass fiber/resin composite have been reported. The present study does not fit the focus on this journal, in which the present investigation does not sufficiently provide highly originality in polymer science, which can contribute to the further progress of this field. In addition, editing of English language and style are required. The quality of figures is bad. The sample as 5%GF/20%NF is presented in Table 6. The same name of sample is presented in Table 3.

Author Response

Comment 2.1:

In this manuscript, the laminated polyester thermoset composites based on palm tree fibers extracted from the palms leaflets and glass mats fibers were manufactured to develop hybrid compositions. However, many researches regarding palm fiber/glass fiber/resin composite have been reported.

Response to comment 2.1:

Thank you for this comment. The authors agree that hybrid composites of plant fibers/glass fibers have been studied in literature. Among the studied plant fibers there are the oil palm fibers and suger palm fiber. But at the best of the authors’ knowledge no hybrid composites based on date palm tree/glass fibers have been studied, especially with polyester thermoset resin. On the other hand, in the present paper the originality lies on the combination at the same time. i) a multitude of characterizations of the structure and the properties of these palm tree based composites. A comparative study between non-destructive (Ultrasonic method) and a destructive method (tensile tests) to mechanically characterize the composites. ii) The presentation of solar aging study to show how the fibers and their nature could improve the solar aging resistance of the polyester thermoset matrix. Among the important results obtained consists on showing that, exposure to natural sunlight deteriorated mechanical resistance of the neat resin after only 60 days, while the composites keep high mechanical resistance 365 days of exposure depending on the composition of the fibers (fractions of glass and natural fibers). The fibers act as stabilizing agents.

Thus we have added in the introduction the following paragraph in the introduction:

At the best of the authors’ knowledge no hybrid composites based on date palm tree/glass fibers have been studied, especially with polyester thermoset resin. On the other hand, in the present paper the originality lies on the combination at the same time. i) a multitude of characterizations of the structure and the properties of these palm tree based composites. A comparative study between non-destructive (Ultrasonic method) and a destructive method (tensile tests) to mechanically characterize the composites will be made. ii) The presentation of solar aging study to show how the fibers and their nature could improve the solar aging resistance of the polyester thermoset matrix. A stabilizing effect of the fibers will be explored.

Comment 2.2:

The present study does not fit the focus on this journal, in which the present investigation does not sufficiently provide highly originality in polymer science, which can contribute to the further progress of this field.

Response to comment 2.2:

We Thanks the reviewer for his comment, however to answer to this part, we took a look to the journal scope and it we believe that the paper fits with at list three points in the journal scope list: Polymer Chemistry, Polymer Processing and Engineering, Polymer Composites and Nanocomposites.

Comment 2.3:

In addition, editing of English language and style are required.

Response to comment 2.3:

We are sorry for that. Editing has been done to improve the paper style.

Comment 2.4:

The quality of figures is bad.

Response to comment 2.4:

Thank you for this comment. We agree that the legend of Figure 6 is not very good. We have made a change on this figure to upgrade its clarity.

Comment 2.5:

The sample as 5%GF/20%NF is presented in Table 6. The same name of sample is presented in Table 3.

Response to comment 2.5:

Thank you for this remark; we rectified it in table 6.

Reviewer 3 Report

In recent years, natural fiber reinforced polymer composites have gained much attention over synthetic fiber composites because of their many advantages such as low-cost, light in weight, non-toxic, non-abrasive, and biodegradable properties. On the contrary, natural fibres have some disadvantages, such as high moisture absorption due to repelling nature, low wettability, low thermal stability, and quality variation which lead to the degradation of composite properties. Nowadays natural fibersreinforced polymer composite have greatly increased in the exterior applications. The two dimensional structures have been used in maritime craft, air craft, high performance automobiles, and civil infrastructure. Some of the Natural  fibers advantages are there exceptional properties compared to synthetic ones, such as low cost, recyclability, renewability and good mechanical properties. Despite their use as short reinforcement in polymer matrix, long fiber sheet in composite presents several  advantages when it provides inherent reinforcement in multiple directions  especially when they are suitable for polymer reinforcement. Such studies encourage the use of natural fibers at hybrid composite with synthetic fibers, and this combination has a good effect on development of the mechanical proprieties. In this article, laminated polyester thermoset composites based on palm tree fibers extracted from the palms leaflets and glass mats fibers were manufactured to develop hybrid compositions with good mechanical properties, the mixture of fibers was elaborate to not exceed 25.vol%. Samples were prepared with resin transfer molding (RTM) method and mechanically characterized using tensile and flexural test, hardness, impact test and the ultrasonic waves as a non-destructive technique. Although the topic in this work was interesting, the presentation in this manuscript was very poor. This manuscript should be rejected for published in Polymers. However, if the authors are willing to make the substantial revisions according to my comments, I would be glad to re-review this manuscript. Here are my detailed comments:

  1. The detailed literature review indicates efforts made by the authors. The coherence of the related work, however, is still not clear. It may help the authors by answering the following questions: Why are these works relevant? Which specific problems were addressed? How are the previous results related with the latest work? What are the outstanding, unresolved, research issues? Which of them has been solved by the proposed study? Answering the questions leads to the novelty of the proposed work naturally. Besides, the current one is nothing but a literature review. Why their work is important comparing to previous reports? I think this is essential to keep the interest of the reader.
  2. Materials and Methods part, Although the results look “making sense”, the current form reads like a simple lab report. The authors should dig deeper in the results by presenting some in-depth discussion.
  3. Natural fibers reinforced polymer composites were first introduced in 1908, by combining lignocellulosic fibers with phenolic resin. Natural fibers reinforced polymer composites have been widely used in many fields of life. In the theoretic perspective, fractal theory, analytical model and resin transfer molding (RTM) method are very important tools, which can be used to investigate the mechanical properties of natural fiber reinforced polymer composites (see [Powder Technology, 2019, 349:92-98; International Journal of Hydrogen Energy, 2018, 43(37):17880-17888]). Authors should introduce some related knowledge to readers. I think this is essential to keep the interest of the reader.
  4. In Figs. 4, 5 and 9, the authors should give the explanations for the difference of data collected from different sources.
  5. Results have shown that the use of glass fibers has increased significantly the whole properties; however an optimum combination of mixture could be interesting and could be developed with less glass sheet and more natural fibers which is the goal of this study. On the other hand, exposure to natural sunlight deteriorated mechanical resistance of the neat resin after only 60 days, while the composites keep high mechanical resistance 365 days of exposure. The authors should give some explanation on above conclusions.
  6. Please, expand the conclusions in relation to the specific goals and the future work.
  7. Please expand the motivation, the problem context, clarify the problem description, and (if possible) add specific objectives.
  8. There are also some grammar issues in the text. The authors are required to make a revision throughout the whole manuscript to improve the English writing thoroughly and carefully.

Author Response

In recent years, natural fiber reinforced polymer composites have gained much attention over synthetic fiber composites because of their many advantages such as low-cost, light in weight, non-toxic, non-abrasive, and biodegradable properties. On the contrary, natural fibres have some disadvantages, such as high moisture absorption due to repelling nature, low wettability, low thermal stability, and quality variation which lead to the degradation of composite properties. Nowadays natural fibersreinforced polymer composite have greatly increased in the exterior applications. The two dimensional structures have been used in maritime craft, air craft, high performance automobiles, and civil infrastructure. Some of the Natural fibers advantages are there exceptional properties compared to synthetic ones, such as low cost, recyclability, renewability and good mechanical properties. Despite their use as short reinforcement in polymer matrix, long fiber sheet in composite presents several advantages when it provides inherent reinforcement in multiple directions  especially when they are suitable for polymer reinforcement. Such studies encourage the use of natural fibers at hybrid composite with synthetic fibers, and this combination has a good effect on development of the mechanical proprieties. In this article, laminated polyester thermoset composites based on palm tree fibers extracted from the palms leaflets and glass mats fibers were manufactured to develop hybrid compositions with good mechanical properties, the mixture of fibers was elaborate to not exceed 25.vol%. Samples were prepared with resin transfer molding (RTM) method and mechanically characterized using tensile and flexural test, hardness, impact test and the ultrasonic waves as a non-destructive technique. Although the topic in this work was interesting, the presentation in this manuscript was very poor. This manuscript should be rejected for published in Polymers. However, if the authors are willing to make the substantial revisions according to my comments, I would be glad to re-review this manuscript. 

Response to comment 3:

Thank you to the respectable reviewer for this introduction, which shows the importance of conducting this study. The authors will try to response the reviewer comments to confirm that they believe that this study is worthy to be conducted and published.

Here are my detailed comments:

  1. The detailed literature review indicates efforts made by the authors. The coherence of the related work, however, is still not clear. It may help the authors by answering the following questions:

Why are these works relevant? Which specific problems were addressed? How are the previous results related with the latest work? What are the outstanding, unresolved, research issues?

Response to comment:

First We thank the reviewer for his comment, and for this presented work, we were presenting a large study from development of composites by resin transfer molding (RTM) method, optimizing of RTM parameters to get a visually good hybrid composites to characterizing the manufactured composites using many techniques (: solar aging, ultrasonic method to get mechanical properties, analytical methods as tensile, flexural, impact,…tests, hardness and microhardness tests to evaluate the composites surfaces after solar irradiation).

The main problem was to find an optimal combination between natural fiber and glass ones, to keep good mechanical properties and less used synthetic fibers.

As the reviewer described earlier, this field have gained much attention over synthetic fiber composites because of their many advantages. To response to your question, we can see from the literature that some founded results are

In accordance with previous results, (we have shown it in the references part), however, as we said earlier we combined many techniques to study the mechanical properties (tensile tests, flexure tests, hardness test, ultrasonic tests). A comparison between destructive and non-destructive techniques presents some originality of the paper. So researches are not much developed in these technics in the polymer composite field. On the other hand the solar aging study enabled to show the stabilising effect of the filler. This might be considered to optimise the amount of UV-stabilizing agent in composites, especially those based on natural fibers.

We added some paragraphs in the introduction to put in advance these points. We are thankful to the reviewer for this comment.

Which of them has been solved by the proposed study?

Response to comment:

In this study we worked on hybrid and we tried to find the optimal properties with the use of less glass fibers. So as result we found that a substitution of 5% glass fiber by 5% natural fibers could be a mechanically interesting material. 5% is representing 20% of the filler, which is a considerable fraction of the filler!

Answering the questions leads to the novelty of the proposed work naturally. Besides, the current one is nothing but a literature review. Why their work is important comparing to previous reports?

I think this is essential to keep the interest of the reader.

Response to comment:

Thank you for this comment, changes have been made in the introduction and in the conclusion of highlight the points discussed above.

2.Materials and Methods part, Although the results look “making sense”, the current form reads like a simple lab report. The authors should dig deeper in the results by presenting some in-depth discussion.

Response to comment:

We add some supplementary information in the results part

3.Natural fibers reinforced polymer composites were first introduced in 1908, by combining lignocellulosic fibers with phenolic resin. Natural fibers reinforced polymer composites have been widely used in many fields of life. In the theoretic perspective, fractal theory, analytical model and resin transfer molding (RTM) method are very important tools, which can be used to investigate the mechanical properties of natural fiber reinforced polymer composites (see [Powder Technology, 2019, 349:92-98; International Journal of Hydrogen Energy, 2018, 43(37):17880-17888]). Authors should introduce some related knowledge to readers. I think this is essential to keep the interest of the reader.

Response to comment:

We tried to introduce some knowledge to readers in the text.

4. In Figs. 4, 5 and 9, the authors should give the explanations for the difference of data collected from different sources.

Response to comment:

We add some references and data collected from other researches, to explain more the figure 5 and its utility, however the obtained results in these figure 4, and 5 still common in whole composites researches.

For figure 9, it presents a new characterisation (solar aging vs young’s modulus by ultrasonic measurement). So there are no references to compare with in the literature

5. Results have shown that the use of glass fibers has increased significantly the whole properties; however an optimum combination of mixture could be interesting and could be developed with less glass sheet and more natural fibers which is the goal of this study. On the other hand, exposure to natural sunlight deteriorated mechanical resistance of the neat resin after only 60 days, while the composites keep high mechanical resistance 365 days of exposure. The authors should give some explanation on above conclusions.

Response to comment:

It is normal to keep good properties after solar aging in composites more than in the neat resin. This is common when adding filler charges. However, micro hardness test showed that after 90 days the whole samples are affected by solar degradation.

6. Please, expand the conclusions in relation to the specific goals and the future work.

Response to comment:

Changes have made on the conclusion.

7. Please expand the motivation, the problem context, clarify the problem description, and (if possible) add specific objectives.

Response to comment:

Changes have been made in the introduction as well as in different parts of the paper. We hope that these changes have clarify the weaknesses noticed by the reviewer.

8. There are also some grammar issues in the text. The authors are required to make a revision throughout the whole manuscript to improve the English writing thoroughly and carefully

Response to comment:

We are sorry for that. Editing has been performed on the paper.

Round 2

Reviewer 1 Report

no additional comments

Reviewer 2 Report

Accept in present form

Reviewer 3 Report

It is ok.